# Compressive sensing based secure data aggregation scheme for IoT based WSN applications

**Ahmed Salim[1,5]**, **Ahmed Ismail** [3,6]*, **Walid Osamy[2,3]**, **Ahmed M. Khedr[4,5]**

**1** Dept. of Computer Science, College of Science and Arts, Al-methnab, Qassim University, Buridah, Al-mithnab, KSA, **2** Dept. of Applied Natural Science, Applied College in Unaizah, Qassim University, Unaizah, KSA, **3** Computer Science Dept., Faculty of Computers and Artificial Intelligence, Benha University, Benha, Egypt, **4** Computer Science Dept., University of Sharjah, Sharjah, UAE, **5** Mathematics Dept., Faculty of Science, Zagazig University, Zagazig, Egypt, **6** Tashkent State University of Economics, Tashkent, Uzbekistan

☉ These authors contributed equally to this work.
* ahmed.aziz@fci.bu.edu.eg, aziz.bfci@gmail.com

**Data Availability Statement:** All relevant data are within the paper.

**Funding:** The authors gratefully acknowledge Qassim University, represented by the Deanship of Scientific Research, on the financial support for

## Abstract

Compressive Sensing (CS) based data collection schemes are found to be effective in enhancing the data collection performance and lifetime of IoT based WSNs. However, they face major challenges related to key distribution and adversary attacks in hostile and complex network deployments. As a result, such schemes cannot effectively ensure the security of data. Towards the goal of providing high security and efficiency in data collection performance of IoT based WSNs, we propose a new security scheme that amalgamates the advantages of CS and Elliptic Curve Cryptography (ECC). We present an efficient algorithms to enhance the security and efficiency of CS based data collection in IoT-based WSNs. The proposed scheme operates in five main phases, namely Key Generation, CS-Key Exchange, Data Compression with CS Encryption, Data Aggregation and Encryption with ECC algorithm, and CS Key Re-generation. It considers the benefits of ECC as public key algorithm and CS as encryption and compression method to provide security as well as energy efficiency for cluster based WSNs. Also, it solves the CS- Encryption key distribution problem by introducing a new key sharing method that enables secure exchange of pseudo-random key between the BS and the nodes in a simple way. In addition, a new method is introduced to safeguard the CS scheme from potential security attacks. The efficiency of our proposed technique in terms of security, energy consumption and network lifetime is proved through simulation analysis.

## 1 Introduction

The evolution of IoT technology allows for networking and connectivity to billions of items, data, people, and applications. The capability of objects to communicate and collaborate in network opens the door to new innovations in different application domains. Moreover, in

this research under the number (mcs-as-2020-1-3-I-10160) during the academic year 1441 AH /2020 AD. The funders had no role in study design, data collection and analysis, decision to publish, or preparation of the manuscript.

**Competing interests:** The authors have declared that no competing interests exist.

recent years, things have become major traffic generators and receivers over the Internet, with the increasing count of internet connections by things over humans [1–3]. Most of IoT devices engage in wireless communication with each other or with the Base Station (BS), and as a result, Wireless Sensor Networks (WSN) have become one of the most important constituent that enables the IoT model. Integration of WSN devices and other IoT enabling technologies lead the way for future innovations in various sectors including environmental conservation, green applications, etc. [4–9]. IoT based WSN devices are mainly responsible for sensing and sending the collected data to the BS where it will be further processed. Such data will often be sensitive and convey private information as well (e.g. e-health). Therefore, security and privacy are crucial in IoT-based applications. Encrypting data sent between the BS and the WSN nodes can improve security, however, it is challenging to implement on resource-constrained WSNs [10]. There are mainly two kinds of encryption algorithms. One is Private-key based, while the other is Public-key based. Even though public-key based encryption techniques (e.g. ECC, RSA, etc.) offer higher security [11], they are not preferred for use on resource-constrained WSN/IoT devices. The encryption strategy based on the private key (e.g. AES, DES, etc.) doesn't need high computation power and memory [12], however, such keys are required to be pre-stored in nodes which are susceptible to be compromised when kept in unattended environments [13]. Another important concern regarding IoT devices is energy. IoT devices spent the major portion of their energy on wireless radio communication. Therefore, ensuring security and enhancing the lifetime of energy-constrained WSN/IoT simultaneously is a challenging problem.

CS is considered as a symmetric-key cryptosystem that includes input signal $x$ as plain-text, result sample $y$ as cipher-text, and the measurement matrix as key; Also the, measurement process as encryption function, and reconstruction process as decryption function. According to CS theory, signals can be sampled at a sub-Nyquist rate (that is, at a rate smaller than that of the Nyquist theory of Shannon), where the signal can be sparse in nature or sparsified with respect to a basis. Using CS, a signal can be successfully recovered using samples, without any significant loss in the information. Moreover, in contrast to traditional compression methods, CS allows simultaneous sampling and compression, contributing to energy-efficiency [5, 8]. Compared to non-CS-based methods (the symmetric-key cryptosystem), the CS as a cryptosystem cannot be used for universal encryption. The general cryptosystems are not able to be quantized. However, the output of the CS can be quantized since robustness of CS can still guarantee a feasible signal reconstruction. Moreover, CS cryptosystem has the extra benefit of compression capability, i.e., it encrypts and compresses the data in one step [8].

To achieve high security and performance efficiency in IoT based WSN data collection, we present an efficient security scheme with the following objectives: As with any private key algorithms, CS-based encryption method requires that the two communication parties (sender and receiver) must agree and know the CS matrix. The sender will use it to encrypt and compress its data and the receiver will use it as an input for any CS reconstruction algorithm to reconstruct the original data. However, like any private key algorithm, CS also suffers from key sharing challenges where the sender and the receiver have to share the key through an insecure channel, intensifying the possibility of different security attacks. Elliptic-curve cryptography (ECC) [14] is a public-key cryptographic technique built on the algebraic structure of elliptic curves over finite fields. The ECC algorithm generates both public and private keys, which makes the encrypted data more safe (please refer to section 3.3 for ECC background). Public key algorithm such as ECC can be a choice to perform encryption at the node side and to provide data protection. However, this would cause Cipher-Text Expansion (the increment of the message length after it got encrypted, given by dividing the Plain-Text size by the Cipher-Text size), which has direct impact on increasing the communication load on resource

constrained WSN nodes. Therefore, it is required to compress node data before encryption to reduce communication load; CS has such powerful properties as it can compress and work as a private key encryption algorithm simultaneously where the CS compression equation (Eq 2 is considered as encryption equation with plain-text $x$, cipher-text $y$, and secret private key the CS matrix $\Phi$).

## 1.1 Motivation and contribution

Inspired by what have been discussed above, we provide a solution by integrating CS and ECC. We consider the benefits of each of them and devise a security technique for CS based data collection in IoT/WSNs, which provides security as well as energy efficiency for cluster-based WSNs. This can be achieved by lowering the data dimension using compressive data sampling in addition to securing the CS measurement matrix. A new key sharing method is introduced to exchange the pseudo-random key securely between the BS and the nodes in a simple way. It helps in solving the aggregation issue without requiring the private key at the Cluster Head (CH) side. Moreover, we introduce a new method to safeguard the CS scheme from the potential security attacks. The performance efficiency of the proposed technique is evaluated through simulation analysis. The contributions of the proposed technique can be summarized as follows:

- Improve security by introducing a technique that generates and exchanges the key between the WSN nodes and the BS in a simple and secure manner. Every WSN node uses a simple chaotic map and generates a random number, whereas two random numbers are generated by the BS ($g_s$ and $e_2$): the first one ($g_s$) acts as the CS matrix seed which should be shared by both sides to encrypt and decrypt sensors data and the second one ($e_2$) is used by the BS to perform the sharing process during the proposed Key Exchange algorithm. This scenario would be used by both parties only once before any transmission.

- Provide security enhancement by withstanding CS-based security attacks using the newly introduced algorithm: Data Compression with Encryption", which allows the WSN nodes to use their secret value to generate secret compressed samples.

- Utilize the security performance of the ECC algorithm against CS attacks and integrate it with CS method during the transmission process between the CHs and the BS. By this approach, our technique achieves high security performance against the possible attacks with low energy consumption.

- Improve security by solving the security threats facing CS with the help of the proposed Key Regeneration algorithm with which the BS and the WSN nodes regenerate the CS matrix seed dynamically and independently in every iteration.

The paper is further structured as follows: Section 2 discusses the related work. Section 3 provides the background study on: CS, Different potential attacks facing CS-based IoT networks, and ECC. Section 4 explains the proposed security solution in detail. In Section 4.8, an example scenario is given. In Section 6, we provide the performance results of our method and compare it with the baseline algorithms. In Section 7, we conclude the paper.

## 2 Related work

Both data privacy and security have equal significance in wireless networks and IoT technologies [15] since a wide variety of vital applications (e.g. transportation) rely on low power and low data connectivity sensors to send important as well as confidential information [1, 16–18].

In Vehicular ad hoc network (VANET) environments, the security and intelligent decision making are two important challenges [19–25]. In [19], the authors proposed a trusted authority (TA) to provide a variety of online premium services to customers through VANETs and to maintain the confidentiality and authentication of messages exchanged between the TA and the VANET nodes. In [20], an efficient anonymous authentication scheme to avoid malicious vehicles entering into the VANET is proposed. In addition, the proposed scheme offers a conditional tracking mechanism to trace the vehicles or roadside units that abuse the VANET. In [21], a novel approach to improve the existing authentication support to VANETs. In this proposed framework, first an anonymous authentication approach for preserving the privacy is proposed which not only performs the vehicle user's anonymous authentication but preserves the message integrity of the transmitting messages as well. In [22], a novel anonymous mutual and batch authentication schemes for improving VANET security is proposed. The proposed scheme makes use of some well-known cryptographic operations to authenticate vehicles. In this scheme, the vehicle users communicate with Road Side Units (RSUs) to get location based information (LBI) to enrich their driving comfort. The security strength of the proposed scheme is analyzed against the various security attacks to aid a better performance than the previously reported schemes. In [23], an efficient batch authentication and key exchange schemes are proposed to provide a high level security by evading communication with the malicious vehicle users. In [24], due to the decentralized nature of blockchain technology, rapid reauthentication of vehicles is achieved through secure authentication code transfer between the consecutive roadside units. security strength of the proposed blockchain-based anonymous authentication scheme against various harmful security attacks is proven. In addition, blockchain is used to substantially diminish the computational cost compared to conventional authentication schemes.

Normally, a data encryption scheme can provide security between the IoT parties. Based on the encryption scheme, a lot of researches has been proposed [26–32]. The authors of [26] proposed an ECC and Diffie-Hellman based method that can be applied to different levels of the network. In [27], authors proposed a confidentiality and integrity algorithm that uses homomorphic encryption with a symmetric key for protecting data privacy. Suganthi et al. [28] presented a key-management strategy that uses three categories of keys shared by every sensor node. Kadri et al. [29] proposed an algorithm that uses a symmetric key between sensors and BS. It uses multi-hop transmission targeted to achieve minimum energy consumption, better scalability, and high security. In [30], the authors presented a symmetric key cryptographic scheme for hierarchical clustered WSN which uses single-hop transmission. The main target of the scheme is to reduce the probability of eavesdropping. In [33], the BS and the sensor nodes produce identical CS matrix in each round, rendering it susceptible to the Known Plain-text Attack (KPA). In [31], the authors proposed a symmetric key algorithm to reduce the number of operations per round and time. In this approach, some steps of the round function are merged and blended by randomly generated mixing bijection. The authors of [32] proposed a symmetric key-based cryptographic technique using Cellular Automata Rules (CA Rules) to encrypt and decrypt sensor data. All the above algorithms were able to provide data privacy and security, but, due to their high computational complexity, they cannot be considered as a security solution for IoT devices with limited power and storage.

In the recent years, secure data collection by solving both the energy and security challenges have been investigated by the research community. CS based data collection schemes with the ability of simultaneous compression and encryption have been utilized for reducing data collection cost and improving the lifetime performance of the network.

A lot of work has been proposed (e.g., [5, 13, 33–50] by utilizing CS as a security scheme regardless of its security degree to achieve data privacy, security, energy and efficiency.

A cluster-based CS routing method to minimize the consumed energy by exploiting the temporal and spatial correlation, called EECSR, is presented in [44]. Semi-variance based CS (SCS) algorithm is proposed in [49]. Based on Spatio-temporal correlations between measurements, SCS gathers samples from nodes to monitor climatic data, and the use of spatio-temporal sparsification helped to reduce the energy usage of nodes that are associated in space and time. [48] introduced a model for energy consumption analysis. Relying on this model, the sources for energy usage in CS based WSNs are grouped into two categories: communication and computation, and are modeled using their components. Cluster-Tree based data routing scheme (CTRS-DG) of [46] includes two layers. Routing, and aggregation and reconstruction. A dynamic and self-organizing entropy-based method of clustering is introduced in the aggregation and reconstruction layer. At CHs, data is aggregated and compressed using the CS scheme.

In [50], an economic theory integrated clustered routing with CS, called EIREC, is proposed to enhance WSN energy efficiency without any recharging equipment. The energy overhead resulting from spatio-temporal correlation is reduced using CS and the energy consumption in inter cluter communication is reduced using a new Energy Efficiency Welfare concept. In [34], a CDG algorithm (Compressive Data Gathering) has been proposed, that uses the CS method for data gathering in large-scale networks. Each node adopts the global seed to perform encryption and compression of its data. This seed is updated in each round by the BS to change the CS matrix.

The schemes in [5, 44–50] adopt CS to save resources and extend the lifespan of WSNs, but they cannot offer good performance in providing security.

In [51], a CS-based security strategy for data collection (SeDC) is proposed, where the authors integrated CS and public key algorithms to achieve a high-security level. However, performing computations such as encryption and compression besides the public key size at each node leads to a decrease in the network lifetime. El Gamal based sparse compressive data gathering (ESCDG) [43] is proposed with the objective to improve the performance of CDG by utilizing the sparsity of the perception matrix. ESCDG combines El Gamal encryption algorithm and sparse random matrix-based compressed sensing technology for secure data collection and reduced resource utilization in WSN.

In [35], the authors improved the security level of CDG by proposing Secure CDG (SCDG). In each round of SCDG, the BS and the WSN nodes generate a global seed using the hash function. However, the communication cost of SCDG is high because, according to the SCDG security mechanism, some information is needed to be shared between BS and sensors in each round,. In [36], the authors proposed an algorithm that uses CS for security. They used random linear projections to generate the compressed samples for use as cipher-text. In [37], the authors used CS to find a solution for authentication and tamper detection problems. The key generation can be accomplished using RSS (Received Signal Strength) based techniques as presented in [38, 39]. However, the generated keys using such techniques are applicable for conventional cryptographic encryption algorithms such as ECC and RSA. Another work presented in [13, 40, 41] used channel measurements for generating keys suitable for CS-based cryptography, which doesn't make use of any strategy for distribution of keys. However, the above techniques cannot be used in IoT because they involve a large number of steps for the key generation, which is difficult to be performed successfully in resource-constrained WSN nodes as it can result in increased power consumption.

A CS aided data acquisition system is proposed in [52]. In this system, CS data is noised randomly to improve the security of data communication. However, owing to the use of a symmetric encryption key, this method has several problems with key management and storage space. In contrast to this, our proposed framework realizes all such complex mathematical

computations for the generation and exchange of keys using BS. An Implementation scheme of Domingo-Ferrer's Homomorphic Encryption for WSNs Integrated with Cloud Infrastructure is introduced in [53]. In [51] an adaptable secure compressive sensing-based data collection scheme for distributed WSN is proposed where both encryption and decryption are used by each node, but they are computationally intensive operations. The proposed algorithm in [54] shows that the decrypted data will be sparser than that of the actual data when an attacker tries to encrypt the data with a wrong encryption matrix. While [55] provides an insight that CS cannot be considered as immaculately secure, [54] demonstrates that the measurement matrix can facilitate secure computations from attacks such as Cipher-text Only Attack (COA) and brute force attack. Even though the encryption methods using CS can offer computational secrecy to withstand attacks such as COA and brute-force attacks, these schemes do not handle the case of CP-Attack (Chosen Plain-Text Attack). The CP-Attack (CPA) scenario was first addressed in [56] where the author used Fractional Fourier Transform (FRFT) as the secret basis for sparsifying. However, the complexity of this method restricted its applicability to power and storage constrained sensor nodes. The authors of [57] proposed another solution to address the CPA, efficient in terms of computation and memory, using chaotic sequences as secret values. However, they didn't provide enough explanation on how the secret values are exchanged securely between legitimate users. In addition to this, [58] also uses a chaotic system to overcome challenges such as low-cost sampling and confidentiality. The authors in [59] proposed a cache decision system that operates over an smart buildings, which will offer the users safer and efficient environment for browsing the Internet, sharing and managing large-scale data in the fog. In [60] data mining classification technique has been used in order to group the connected devices based on the collected data and then detect the nodes which generate erroneous data. A multi-agent-based data collection and aggregation model is proposed for monitoring fog infrastructure in [61]. Secure decentralized spatial crowdsourcing scheme for 6G-Enabled Network is proposed in [62] in which nodes can gather and transmit information on the blockchain without depending on third party. The work presented in [63] introduced a new variant of the optimistic concurrency control protocol for validating the transactions that shall be carried out partially at the fog and globally at the cloud server.

All of the above methods and techniques act as private key algorithms and they suffer from key distribution challenges. Moreover, all of the previous works are vulnerable to KPA attacks because they used a single CS matrix during their encryption and decryption process. In this paper, we propose a CS security scheme for IoT based WSNs. The proposed method integrates between CS-method (as encryption and compression method) and Elliptic-curve cryptography (ECC) (as public key algorithm), such that CS supports to solve the aggregation issue without requiring to store the private key at the CH side, and solves the CS- encryption key distribution problem by enabling the BS and nodes to securely exchange the pseudo-random key in a simple way. A new key sharing algorithm is introduced to address the key distribution challenge. Moreover, the proposed strategy improves security with its ability to withstand CPA and COA using the proposed Data Compression with an Encryption algorithm that allows the nodes to use a secret value to generate secret compressed samples. Finally, the proposed scheme provides resistance against the Known Plain-text Attack (KPA) using Key-Regeneration Algorithm. The notations used are provided in Table 1.

## 3 Background study

In this section, we first provide the background information on Compressive Sensing. Then, we discuss different possible attacks on the CS-based encryption method in IoT networks.

**Table 1. Notations description.**

| Notation | Description |
|---|---|
| $x$ | Sensors readings |
| $g$ | Sparse presentation of $x$ |
| $\Phi$ | Measurement matrix |
| $\xi$ | Global seed |
| $S$ | Sparse level (number of non zeros values) |
| $\Psi$ | Transform matrix |
| $r$ | Number of round |
| $n_j.\alpha$ | Coefficient vector for node $n_j$ |
| $\Theta$ | $M \times N$ matrix such that $\Theta = \Phi\Psi$ |
| $n_j.y$ | Compressed vector for node $n_j$ |
| $y$ | Measurement vector (compressed samples) |
| $ECC$ | Elliptic-Curve Cryptography |
| $Epr$ | ECC Private key |
| $E_{pu}$ | ECC Public key |

Finally, we discuss the ECC algorithm, which is a public-key cryptographic technique built on the algebraic structure of elliptic curves over finite fields.

## 3.1 Compressive Sensing (CS)

The CS method allows sampling and compression to be executed in one step. This differentiates it from conventional techniques of compression where sampling and compression are performed in separate steps [64]. Also, the CS reconstruction technique often requires no prior expertise to retrieve the actual data from the compressed samples successfully [64].

Consider $x[n] \in R^N$ be the reading vector obtained from $N$ sensors, where $n = 1, 2, \ldots, N$. A signal in $R^N$ can be conveyed by utilizing a basis of $N \times 1$ vectors $\{\Psi_i\}_{i=1}^{N}$. Let the basis be orthonormal for simplicity. Therefore, any signal $x$ can be represented as given in Eq 1, where $\Psi$ denotes $N \times N$ orthonormal matrix used for transformation, $\Psi_i$ denotes a column of the matrix, $g$ is an $N \times 1$ matrix to store the sparse presentation of $x$ [64].

$$x = \sum_{i=1}^{N} \Psi_i g_i \quad or \quad x = \Psi g. \tag{1}$$

Here, the CS focuses on signals that are sparse by nature or that can be sparsified with respect to some basis. That means, the signal $x$ has just $S$ basis vectors, $S \ll N$, where, only $S$ components of $g$ are non-zeros and the remaining $(N - S)$ components are zeros.

Applying Eq 1, the compressed samples vector $y \in R^M$ can be obtained using Eq 2:

$$y = \Phi x = \Phi\Psi g = \Theta g. \tag{2}$$

with $M \ll N$, and $\Theta$ the $M \times N$ matrix.

**3.1.1 CS signal reconstruction process.** Consider the CS scenario which requires to reconstruct a larger and sparse signal using a few available measurements coefficients. One of the easiest solutions to reconstruct the signal from its available measurements using Eq 2 is to find a solution of the $\|L\|_0$ minimization issue which determines the non-zero entries count, and the problem of signal reconstruction becomes:

$$x = arg \ min \ \| x \|_0 \quad s.t \quad y = \Phi x \tag{3}$$

Even though this works well theoretically, it is computationally NP-hard [8]. It is computationally hard to determine a solution to the issue (defined by Eq 3) for any vector or matrix. However, the CS framework provides efficient alternate solutions to Eq 2 by using Basic Pursuit (BP) [65] or Greedy Pursuit (GP). Examples of Greedy Pursuit includes Orthogonal Matching Pursuit (OMP) [66], ROMP [67, 68], and Stagewise Orthogonal Matching Pursuit (StOMP) [69].

## 3.2 Different potential attacks on CS-based IoT networks

The following three potential attacks on the CS-based encrytion method in IoT networks are briefly discussed in this section: Cipher-text-Only attack(COA) [70], Chosen-Plain-text Attack (CPA) [56] and Known Plain-text Attack (KPA) [52].

**3.2.1 Ciphertext-only attack (COA).** In COA, the attacker is assumed to have access to only a limited set of cipher-texts. The attack is said to be successful if the matching plain-texts can be deduced/extracted. A success is described as the ability to extract any amount of information from the underlying cipher-text.

**3.2.2 Chosen-plaintext attack (CPA).** In the CPA attack, the attacker uses his knowledge of the cipher data $y$ and the original data $x$ (by measurement) and tries to track the wireless link to obtain the private key $\Phi$. Also, the node's data $x$ can be altered by the CPA attack in various ways, such as choosing one node as a CPA target node, and then the noise value for the target node is boosted by attaching a source of radiation close to the node. Therefore, in many situations, a CPA attack is possible. The goal of the attacker is to identify the secret key $\Phi$ from $x$ to $y$. For this, he may repeat the attack several times, to obtain N distinct values of $(x, y)$. Then by solving the least-squares method he can recover the private key $\Phi$.

$$\Phi' = arg_{min}\Phi \parallel y - \Phi x \parallel_2 \tag{4}$$

**3.2.3 Known plain-text attack (KPA).** In the Known Plain-text Attack (KPA), it is assumed that both the cipher-text and the underlying plain-text are known by the attacker and that he tries to acquire the CS matrix (i.e., the private key). Thus, KPA focuses to derive the $\Phi$ (i.e. the key), the measurement matrix using the known measurement $y$ (the cipher-text) and the sensing matrix $x$ (the plain-text). In many situations, KPA are feasible at an acceptable cost, making CS encryption-based systems vulnerable to this attack. In this type of attack, multiple attempts made by the attacker let him to obtain at least $k$ pairs of plain-text and the associated $(p, q)$ cipher-text. For example, assume that the attacker succeeds to obtain Plain-text $p_j = (p_1, p_2, p_3, \ldots, p_k)$ and associated Cipher-text $q_j = (q_1, q_2, \ldots, q_k)$ after repeating his attempt $j$ rounds. Then, he can obtain the new linear equation given by $q = \Phi p$ and can very well retrieve the $\Phi$ matrix (i.e., the secret key) by solving the least-squares problem.

## 3.3 Elliptic-curve cryptography (ECC)

ECC [15] is a public-key cryptographic technique built on the algebraic structure of elliptic curves over finite fields. The ECC algorithm generates both public and private keys, which makes the encrypted data more safe. In the public-key cryptography domain, ECC exhibit better performance than other public-key protocols such as RSA in terms of power usage, key sizes, memory and throughput. As a result, ECC provides a robust solution for resource-critical applications in terms of data transfer confidentiality, data reliability and authenticity, and non-repudiation, especially in the wireless communication system. The general equation of the elliptic curve can be expressed as follows (Eq 5):

$$y^2 = x^3 + Ax + B \tag{5}$$

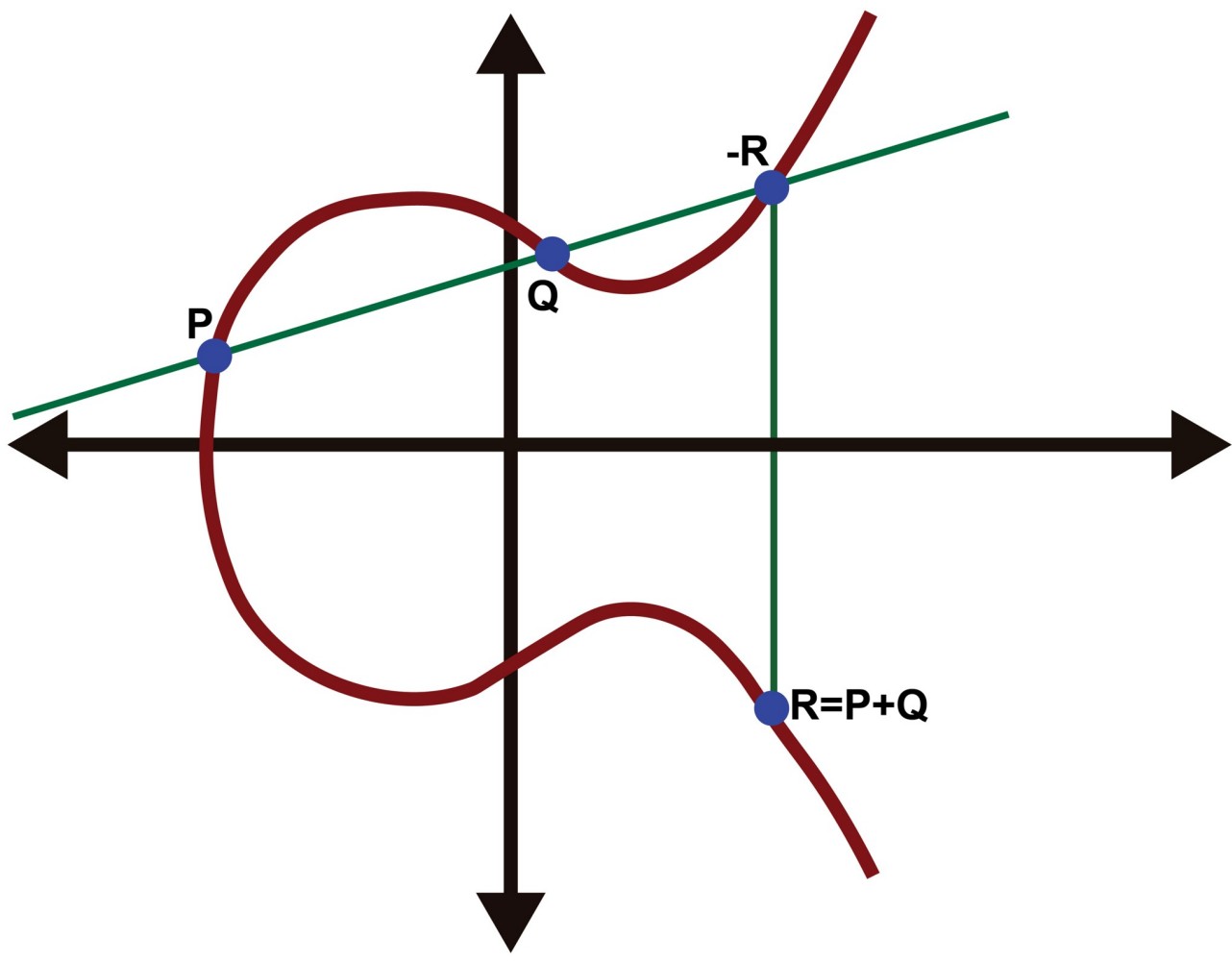

**Fig 1. ECC Curve [15].**

where *A* and *B* are constant. Taking a point on the curve (Fig 1) and multiplying it by a number yields another point on the curve. Even though you know the original point and the output, deciding what number was used is challenging. Elliptic curve based equations have a very useful property for cryptography: they are quite simple to execute but incredibly difficult to reverse. For example, in an elliptic curve, consider two points on the curve, *P* and *Q*. It is easy to combine *P* to obtain *Q* as *nP* = *Q*. However, it is very difficult to obtain *P* from *Q*. That is, given a point *Q*, finding the number of times *P* is combined with itself to obtain *Q* is hard. Finding *n*, when *Q* is given, is also impossible in a reasonable amount of time. The ECC algorithm consists of three steps: key generation, encryption, and decryption. The three steps can be expressed as follows:

**3.3.1 ECC key generation.** Key generation is the crucial aspect where ECC generates both private and public keys. The public key will be shared with the nodes for use in encrypting its data and the private key will be used only by the BS to decrypt the data. The key generation process can be summarized as follows:

- Select the random prime number as private key $E_{pr}$

• Generate the public key $E_{pu} = E_{pr} * p$, where $p$ is a point on the curve $E$.

**3.3.2 ECC encryption process.** Let $D$ be the message sent by sensor nodes. This message needs to be represented on the curve. Consider the scenario where $D$ has the point $p1$ on the curve $E$. Then, select $q$ randomly from $[1–(n − 1)]$. $C1$ and $C2$ denotes the two cipher-texts generated using Eq 6

$$C1 = q * p$$
$$C2 = D + q * E_{pu}$$

(6)

**3.3.3 ECC decryption process.** After receiving the encrypted data, the BS will use the Eq 7 to get back the message $D$,

$$D = C2 − Epr * C1$$

(7)

## 4 Proposed security scheme for IoT-based WSNs

Here, we discuss the proposed CS security technique for IoT-based WSNs that integrates between CS-method (as encryption and compression method) and Elliptic-curve cryptography (ECC) (as public key algorithm). The use of CS helps to solve the aggregation issue without requiring the private key at the CH side. The proposed method solves the CS- encryption key distribution problem by enabling the BS and nodes to securely exchange the pseudo-random key in a simple way. Along with this, we introduce a new method to safeguard the CS scheme from all potential attacks. We first discuss the proposed system model and the problem statement. Then, we explain the design of the proposed technique and its different phases of operation in detail.

### 4.1 System model

Depending on the application, the WSN gather environmental data from the surrounding region. The selection of suitable routing protocol for data propagation is important because of its significance in optimizing the energy consumption in WSNs. The hierarchical cluster-based routing protocol is considered the most efficient protocol in terms of scalability and energy efficiency in WSNs [4]. There are several clustering protocols that have been specifically designed for WSNs(e.g., [4–6]). In this paper, we assume that WSN is split into several clusters. For each cluster, one unique node serves as the CH, while the other nodes behave as cluster members (CMs). CHs may be chosen by the cluster nodes or allocated by the network designer. CH gathers and compresses the data sent by the CMs, and the processed data is then transmitted to BS. In this way, the total data amount transmitted to the BS can be significantly reduced.

### 4.2 Problem statement

As discussed earlier, CS strategy has the ability to reduce the data size without involving complex mathematical computations and this makes CS a convenient solution for IoT data processing. Also, its potential to perform simultaneous compression and encryption makes it more attractive. However, depending only on CS-based encryption as a security method is not a good solution. Like any private key algorithm, CS method also suffers from a number of issues, including key distribution and sharing challenges. Although public key algorithms can

alleviate these difficulties, their usage in WSNs has the following limitations, making them an inappropriate option.

- Each node generates two keys: (i) public and (ii) private. A node sends its public key to allow others to communicate with it, while it utilizes the private key for decryption purpose. This process is very complex and consumes huge energy. Hence it is not a wise solution to apply on low-powered WSN devices.

- BS generates the both public and private keys and then transmits the public one to the entire network so that each node can encrypt its data. But in order to aggregate the data at the CH side, the CH should first decrypt, then aggregate and encrypt again for transmission to the BS. This cause the CHs to consume more energy and also require that the CH should know the private key. This makes the employment of public-key algorithm insecure.

To address these challenges, we introduce a CS Security Scheme for IoT-based WSNs with the following characteristics as given below:

- Integration of CS-method (encryption and compression) and Elliptic-curve cryptography (public key algorithm) such that CS can support to solve the aggregation issue without requiring the private key at the CH side.

- Solves the CS- Encryption key distribution problem by introducing a new key sharing method that enables BS and nodes to securely exchange the pseudo-random key in a simple way.

- Introduces a new method to safeguard the CS scheme from the potential security attacks.

We consider the scenario between one node, the BS, and the attacker (refer Fig 2), for simplicity.

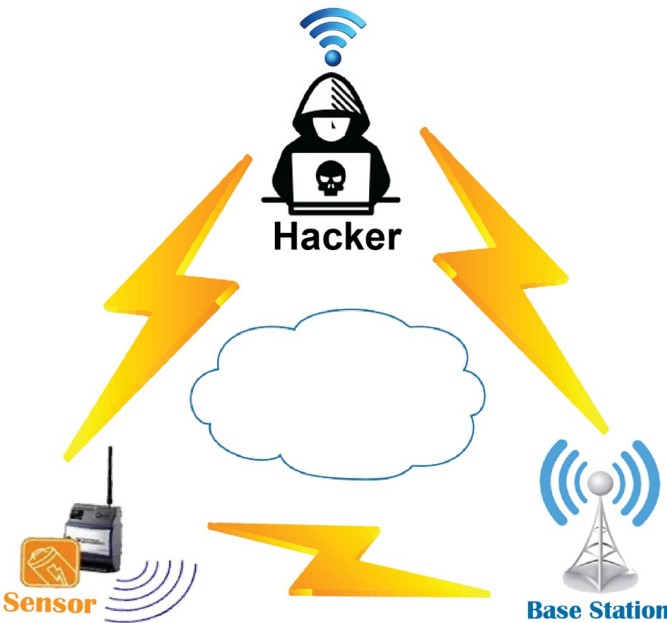

**Fig 2. The proposed scenario: Two legitimate sides (sensor node and BS) and one malicious side (hacker).**

### 4.3 Proposed algorithm design

The proposed scheme has mainly five stages of operation, as follows: (1) Key Generation, (2) CS-Key Exchange, (3) Data Compression with CS Encryption, (4) Data Aggregation and Encryption with ECC algorithm (5) CS Key Re-generation. The first two phases (Key generation and CS-Key exchange) are executed only once in the initial round, while the other three phases (Data Compression with CS Encryption, Data Aggregation and Encryption with ECC algorithm, and CS Key re-generation) are repeated in every round. The flow chart of our CS-Security Scheme for IoT based WSN is given in Fig 3.

### 4.4 Key generation stage

This phase involves the generation of keys at the BS and the sensor nodes. The following are the tasks performed by the WSN nodes and the BS during the key generation process:

- **At the Base Station**: BS generates two types of keys: (i) CS- Matrix and (ii) ECC private and public keys. Firstly, for CS- Key, as mentioned in Section 3, Bernoulli or Gaussian distribution matrix are the most favourable choices. Any technique for pseudo-random number generation uses a vector or a number to start the process (known as seed), which can either be selected randomly or initialized. Therefore, if a node and BS uses identical values for seed, identical random matrix $\Phi$ will be generated for data encryption/ decryption. The main disadvantage of this is that: if the adversary guesses this seed successfully, he/she will be able to produce the same matrix. One goal of the proposed technique is to generate this seed reliably and make it difficult to be guessed by the attacker. We use 1D chaotic maps [57] to generate the seed $g_s$ because we assume there are no resource restrictions at BS. Chaos describes certain non-linear dynamic systems that appear to be random and unpredictable, and we can define the 1-D Logistic mapping equations as follows [57]:

$$c_{n+1} = bd \times c_n \times (1 - c_n). \tag{8}$$

  Here, $bd \neq 0 \in R^+$ is referred to as the biotic potential and every value in Eq 8 is based on the previous value. Eq 8 attains a chaotic-state and produces a chaotic-sequence within the range (0, 1] [57]. Secondly, the BS generates ECC public $E_{pu}$ and private $E_{pr}$ keys as shown in 3.3.1. It keeps the $E_{pr}$ and sends the $E_{pu}$ to each CH.

- **At the WSN node**: CP-attack (CPA) will challenge the CS system by causing the adversary to get the cipher-text $y$ for any plain-text $x$. In order to protect the CS scheme from such an attack, the proposed technique multiplies $y$ with a secret value $S$ and generates the secret compressed sample $y'$. To produce $S$, the node generates a random value $e_1$ and multiplies it with the received seed from BS $(g_s)$, i.e, $S = g_s \times e_1^{-1}$. For generating $e_1$, the node applies simple logistic chaotic map equation [57] given by the quadratic recurrence equation (refer Eq 8).

  The steps performed at a node and the BS can be summarized as follows:

- **At BS**: BS uses Eq 8 to produce $e_2$ and $g_s$. Then generates $E_{pu}$ and $E_{pr}$.

- **At Sensor Node**: Sensor node applies Eq 8 to produce $e_1$ (given by $e_1 = c_{n+1}$), and computes its inverse $e_1^{-1}$ (given by $e_1 * e_1^{-1} = 1$).

### 4.5 CS-Key exchange stage

The CS-based strategy for encryption depends on an assumption that only the BS and the node possess identical values for the seed to produce the same sensing matrix $\Phi$. However, this assumption faces a serious problem if the adversary is listening to their communication

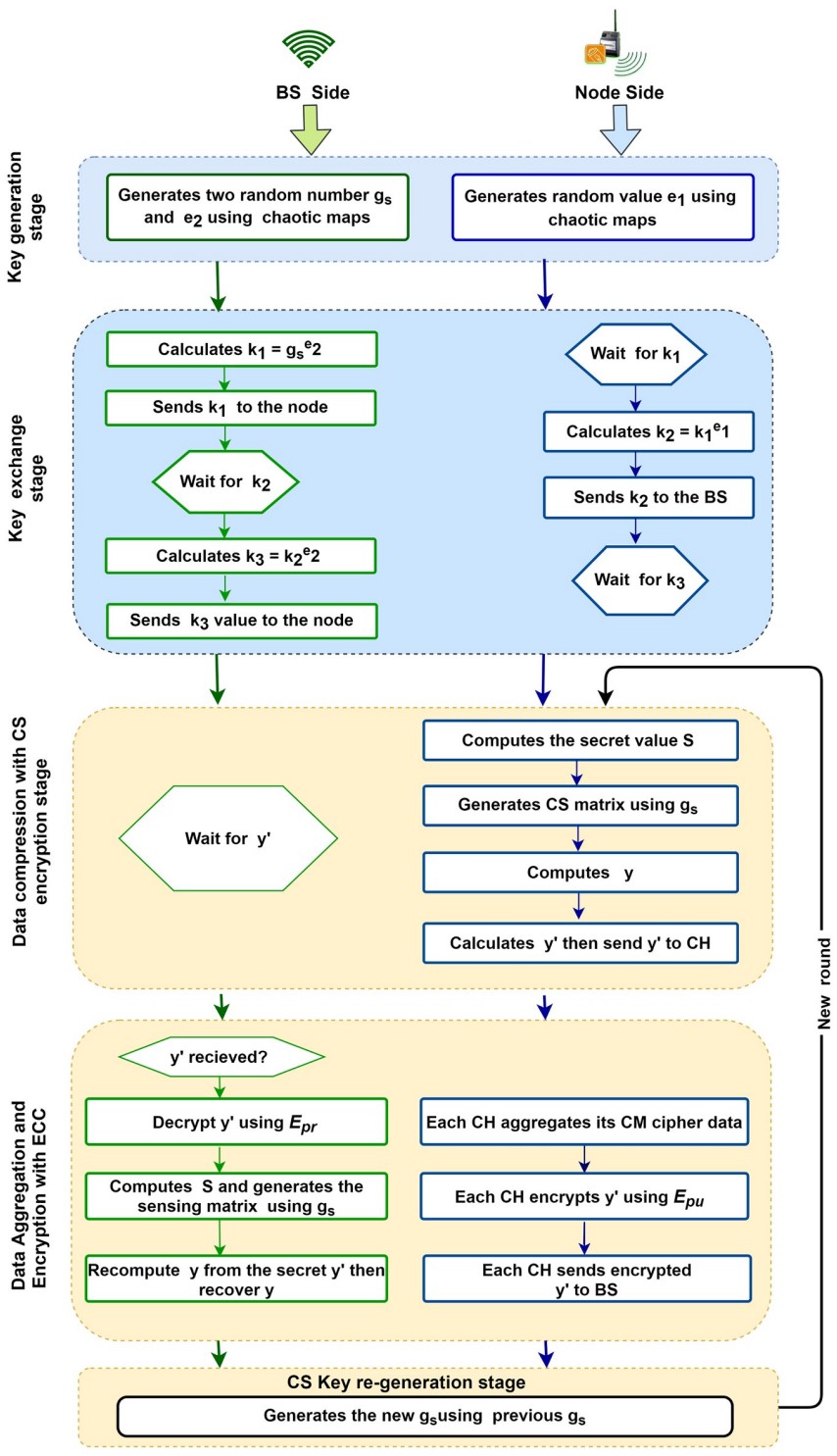

**Fig 3. Flow chart of the proposed scheme.**

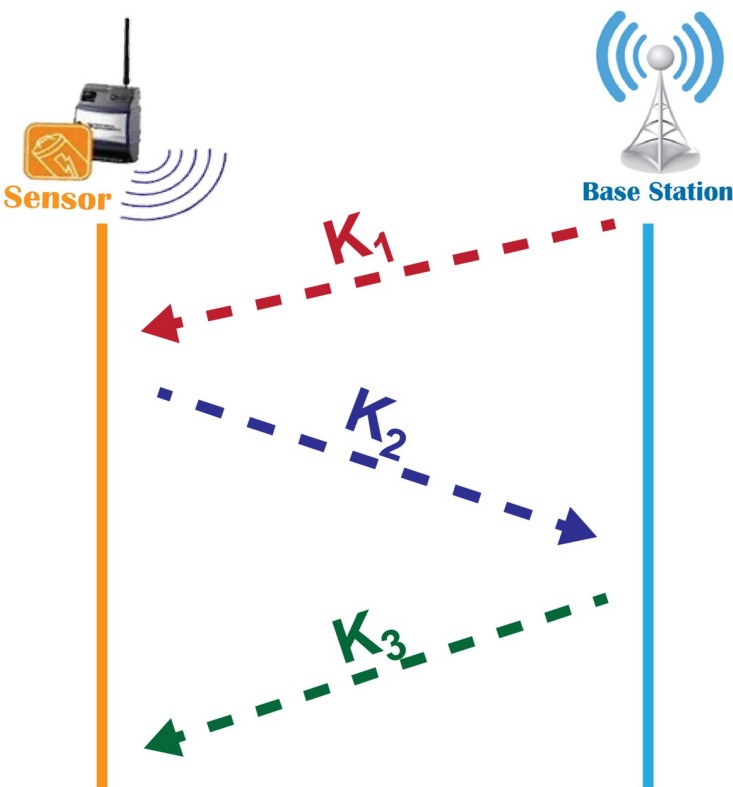

**Fig 4. Procedures for CS-Key exchange phase.**

channel. The nodes and the BS must adopt a strategy to exchange the seed securely. The scenario is illustrated in Fig 2. To prevent this issue, we propose a CS-Key Exchange Algorithm which is presented in Algorithm 1.

The new algorithm permits a safe and simple exchange of the seed between the BS and a node as follows:

- BS computes and sends $k_1 = g_s^{e_2}$ to the node. If adversary obtains $k_1$ and uses it as the seed, the $g_s$ will be hard to guess since BS uses a 1D chaotic map to create this number.

- Then the node computes and sends $k_2 = k_1^{e_1}$ to BS. The adversary will encounter the same difficulty explained above if he/she uses $k_2$.

- The BS calculates $k_3 = k_2^{1/e_2} = g_s^{e_1}$ and sends the value of $k_3$ to the node. The node can generate the seed by computing $g_s' = k_3^{1/e_1} = g_s$.

- Finally, BS and the node possess identical values for seed $g_s$; they initiate the succeeding phase. The steps involved in this phase is shown in Fig 4.

**Algorithm 1**: CS Key Exchange Algorithm

```
1: BS calculates k₁ = gₛᵉ²
2: BS sends k₁ to the node.
3: Node computes k₂ = k₁ᵉ¹.
4: Node transfers k₂ to the BS.
5: BS computes k₃ = k₂^(1/e₂) = gₛᵉ¹.
```

```
6: BS sends k₃ to the node.
7: Node computes g'ₛ = k₃^(1/e₁) = gₛ.
8: Finally, both the node and the BS possess identical values for
g'ₛ = gₛ, and sensing matrix Φ is generated using gₛ as seed.
```

**4.5.1 CS Key exchange algorithm procedure.** Here, we discuss the mathematical steps involved in Algorithm 1. The following steps help nodes and the BS for secure and easy exchange of the seed.

- First, as mentioned in the key generation phase, the node produces $e_1$ and the BS produces $e_2$ and $g_s$ and safely sends $g_s$ to the node.

- The BS computes and sends $k_1 = g_s^{e_2}$ to the node.

- Node uses $k_1$ to compute and sends $k_2 = (g_s^{e_2})^{e_1} = g_s^{e_1 e_2}$ to the BS.

- The BS calculates $k_3 = k_2^{1/e_2} = (g_s^{e_1 e_2})^{1/e_2} = g_s^{e_1}$, then sends $k_3$ to the node

- Finally using $k_3$, the node calculates $g'_s = k_3^{e_1} = (g_s^{e_1})^{1/e_1} = g_s$.

- At the end, the node and BS possess identical values for seed $g'_s = g_s$.

- The BS and the node initiate the next stage.

## 4.6 Data compression with CS-encryption stage

This stage aims to use the CS method in order to compress and encrypt the cluster member's data. In addition, it targets to address the challenges in CS-based encryption and protects it from the potential security attacks. As we discussed above, the node and the BS possess identical values for seed. They use Algorithm 2 for encrypting the compressed data as follows:

- Each node $i$ applies seed to produce the sensing matrix $\Phi_i$ for encrypting the compressed data to $y_i$ using Eq 2.

- Each node $i$ transfers this resultant $y_i$ into secret compressed sample $y'_i$. A secret value $S$ is multiplied with $y_i$ to produce $y'_i$ (refer Eq 9). This step makes the hacker's task more complicated because CPA needs to obtain the same y each time in order to estimate the CS matrix. By using this secret value and one time sensing matrix, the attacker would not be able to get the same $y$ each time.

$$y'_i = y_i \times S. \tag{9}$$

Here, $S = g_s^{-1}$. The node then transfers $y$ to its CH. Now, if a CS-attacker generates any plain-text and send it to the sensor node, the sensor node may encrypt it and sends it back as $y'$ to the attacker. However he/she will generate the wrong cipher-text for the generated plain-text. Hence, the attacker can never generate the correct matrix $\Phi$. Therefore, the proposed scheme is effective in safeguarding the CS scheme from the possible attacks.

## 4.7 Data aggregation and encryption with ECC algorithm

In this stage, the proposed algorithm aims to ensure the security by using the public key as the second encryption mechanism which makes the attacker's mission impossible. The steps performed in this stage can be expressed as follows:

- Each CH aggregates the cipher data $y' = [y'_1, \ldots, y'_{|CM|}]$ of its respective CMs.

- Then, each CH uses the public key $E_{pu}$ to encrypt $y'$ using Eq 6.

- The BS uses $E_{pr}$ to regenerate $y'$ using Eq 7.

- The BS calculates the secret value $S$ given by $S = g_s^{-1}$ and recomputes $y$ from $y'$ (refer Eq 10).

$$y = y'/S. \tag{10}$$

- Finally, the BS applies the same seed $g_s$ to produce $\Phi$ for encrypting and reconstructing the actual data using $y$. This can be achieved with the help of any algorithm for reconstruction, such as OMP algorithm [66]. It is impractical to produce the same $\Phi$ without knowing the seed $g_s$, and thus no one can reproduce $y$ except BS.

**Algorithm 2**: Data Compression with CS-Encryption

```
1: Each node i calculates the S given by S = g_s^{-1}.
2: Each node i applies the seed g'_s to produce Φ_i and then produce y_i.
3: The node computes y'_i given by y'_i = y_i * S
4: Each node i sends y'_i to its CH.
5: Each CH aggregates the cipher data y' = [y'_1,....,y'_{|CM|}] of its respective
   CMs.
6: Each CH sends y' to the BS.
7: BS calculates the secret value S = g_s^{-1}.
8: BS generates Φ by making use of g_s.
9: BS recomputes y such that y' = y/S.
10: Lastly, the BS recovers the actual data with the help of any recon-
    struction algorithm.
```

## 4.8 CS Key re-generation stage

In this phase, the proposed scheme uses a simple and an efficient way to change the CS encryption key (CS matrix) in each iteration without requiring to share any additional information between BS and the nodes. Thus, it will be very difficult to launch an attack such as a Known Plain-text Attack (KPA) to predict the CS matrix. The key idea of this phase is based on the concept of using the random seed to generate the CS matrix. According to Eq 8, $c_n$ is considered as initialization values to generate $c_{n+1}$. Without knowing the value of $c_n$, it will be very difficult to generate the same values of $c_{n+1}$ especially when they are generated using chaotic maps equation, due to the sensitive properties, i.e., a small difference in $c_n$ gives very far values to $c_{n+1}$. After the first iteration, both the node and the BS will possess the same seed $g_s$ and this seed will be used as initiation value ($g_s = c_n$ in Eq 8) by both sides to generate a new seed $c_{n+1} = g_{s_1}$, and then generate a new CS matrix. In the next iteration, both sides will use $g_{s_1}$ as the new initiation value for Eq 8 and generate a new seed $g_{s_2}$, and so on. After $r$ rounds, each node and BS will have an identical value for seed $g_{s_r}$ to encrypt and decrypt respectively, without sharing any information or more complex computation. Algorithm 3 demonstrates the steps performed in this phase.

**Algorithm**: CS Key Re-generation Algorithm

```
1: FOR each Iteration r = 1 to N (given that N is the number of
iterations)
2: Each node and BS use g_{s_{r-1}} as initialization value in Eq 8 to generate
g_{s_r} the new seed.
3: Each node and the BS use the new seed g_{s_r} to generate new key Φ_r.
4: Repeat CS Encryption Phase (see Fig 3).
5: END
```

## 5 Example scenario

In this section, we elucidate our proposed technique by providing an example scenario. Here, we follow the same assumptions as given in Fig 2. A node that want to transfer its $12 \times 1$ sized data $x = [100010001001]^T$ to BS and an adversary is listening to the communication channel. The execution of the developed algorithm will be as follows:

1. **Key Generation Stage**:

   - Using Eq 8, the generated value of $e_1$ by the sensor node will be: $e_1 = 0.5636363626222726$.

   - BS generates two random numbers: $e_2 = 0.635801265819775$ and $g_s = 0.589524844461205$ (using Eq 8).

   - BS generates $E_{pu}$ and $E_{pr}$ and sends $E_{pu}$ to each CH.

2. **CS Key Exchange Stage**: Algorithm 1 is executed by node and BS as follows:

   - The BS calculates $k_1 = g_s^{e_2} = 0.714636102703056$ and then forwards it to the node.

   - The node calculates $k_2 = k_1^{e_2 e_1} = 0.827478985639529$ and then transmits it to the BS.

   - The BS compute $k_3 = k_2^{1/e_2} = 0.742414841804188$ and transmits $k_3$ to the node.

   - Lastly, the node calculates the seed by $g_s' = k_3^{1/e_1} = 0.589524844461205$

   - At the end of this stage, both sides have same values as $g_s = g_s'$.

3. **Data Compression and CS- Encryption Stage**:

   - Using $g_s' = 0.589524844461205$, the measurement matrix $\Phi_{M \times N}$, where $M = 3$ and $N = 12$ by the sensor node will be:
     $\Phi_{3 \times 12} =$

$$
\begin{bmatrix}
0.53 & 0.86 & -0.43 & 2.76 & 0.72 & -0.20 \\
1.83 & 0.318 & 0.34 & -1.34 & -0.06 & -0.124 \\
-2.25 & -1.30 & 3.57 & 3.034 & 0.714 & 1.48
\end{bmatrix}
$$

$$
\begin{bmatrix}
1.409 & -1.20 & 0.48 & -0.303 & 0.88 & -0.809 \\
1.41 & 0.717 & 1.03 & 0.293 & -1.147 & -2.94 \\
0.671 & 1.63 & 0.72 & -0.787 & -1.068 & 1.438
\end{bmatrix}
$$

   - Then, the node calculates the secret value as follows: $S = g_s^{-1} = 1.696281351660333$.

   - Using Eq 2, the measurement $y$ of the sensor node will be: $y = \Phi * x = [0.94246644037;$ $-0.13876101067; 0.62116146902]$

   - The secret compressed sample $y' = y * S = [1.598688247365326; -0.235377714737061;$ $1.053664616268563]$.

   - The node transfers $y'$ to the CH.

4. **Data Aggregation and Encryption with ECC algorithm**:

   - The CH uses the public key $E_{pu}$ to encrypt $y'$ and the encrypted data is transmitted to the BS.

- Then the BS uses the private key $E_{pr}$ to regenerate $y$.

- After that, the secret value at BS will be: $S = g_s^{-1} = 1.045931177538809$ to generate the actual compressed sample $y$ using Eq 10 where, $y = y/S = [0.94246644037; --0.13876101067; 0.62116146902]$.

- Lastly, the BS applies $g_s$ to generate $\Phi$ with the help of any reconstruction algorithm, the BS recomputes $x$ successfully.

5. **CS Key Re-generation Stage**:

- Both node and BS applies the value of $g_s = 0.589524844461205$ as the seed in Eq 8 to generate the new seed $g_{s_1}$.

- Then $g_{s_1}$ will be used by both sides for data encryption and decryption process.

- Repeat this stage in each iteration.

## 6 Simulation results

In this section, we present the performance results of our proposed algorithm, and compare the results with recent existing algorithms. This section is further organized as follows. In section 6.1, we analyze the security feature of the proposed scheme against various statistical inference attacks discussed in section 3.2 and provide the complexity analysis for the proposed scheme in comparison with the other security schemes. Finally, in section 6.2, performance comparison of the proposed technique with other existing algorithms (CDG [34], LSS [33], EIREC [50], SDC [42] and EECSR [44]) in terms of energy efficiency, alive nodes count and network lifetime is given.

### 6.1 Security analysis

This section is divided into two parts: first, the security feature of the proposed scheme against the statistical inference attacks, discussed in section 3.2, is evaluated. Second, the complexity of the proposed security scheme is evaluated and compared with the baseline security schemes.

**6.1.1 Attacks analysis.** As seen before, the proposed security scheme consists of two communication stages: Intra (from CMs to CH) and Inter (from CHs to BS) communication. During the Intra- communication stage, each node uses CS method to encrypt and compress the sensed data for transmission towards CH. During this time, statistical inference attacks can happen and the transmitted data can be obtained using the statistical inference attacks like Brute force, COA, CPA and KPA. To approximate the CS matrix (secret encryption key), the attacker may collect a large number of samples based on the style of the chosen statistical inference attack. In this section, we aim to evaluate the impact of the proposed security scheme against the most well known CS attacks such as brute force attack, CPA, COA and KPA attacks.

- Brute Force Attack: To withstand different types of brute-force model attacks, the key space must be significantly sized. Key space size can be defined as the total count of distinct keys used for encryption. In our work, to produce $e_1$, $e_2$ and $g_s$, the BS and the sensor nodes adopted the 1D chaotic map where precision equivalent to $10^{-14}$ is employed. Therefore, the size of key space reaches $\cong 2^{156}$, which is larger than $2^{128}$, i.e., the key space is relatively larger sized to withstand brute-force type of attacks. Even a very small change in the key used for encryption/decryption itself can produce a different cipher-text/data values. In this test, we aim to show the usefulness of using Chaotic maps in producing the seed and the keys. One

Orginal Image

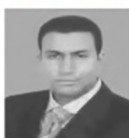

Reconstructed Image (CR)=25%

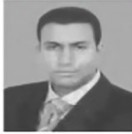

Reconstructed Image (CR=10.0%)  Reconstructed Image (CR=5.0%)

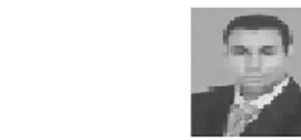

**Fig 5. Decryption process using correct seed.**

of the most important properties of the Chaotic map is that it is very sensitive, i.e., a tiny change in initial value gives different values of random numbers. The proposed scheme uses this advantage of Chaotic map to generate the seed. Even if the attacker generates a number such that the difference between that number and the original seed is so tiny, the attacker will get a different matrix. Hence, he/she won't be able to recover the correct data. To show this, we used the CS method to encrypt and decrypt $512 \times 512$ first author image using the original seed, and then the same seed is used to reconstruct the image. Fig 5 shows that we can successfully reconstruct the original image with different quality depending on the compression ratio (CR), where the lowest reconstruction quality is when CR = 5% and the best reconstruction quality is when CR = 25%. In contrast, if we make a very small change in the original seed ($g_s = g_s - 0.00000000000001$) as shown in Fig 6, we cannot reconstruct the original image.

- CPA and KPA attacks: These types of attacks require to collect a large number of samples to estimate the CS matrix (encryption key). The proposed security scheme depends on One Time Sense (OTS) method to protect the data from these types of attacks [52, 56, 70–72]. That is, the CS matrix is changed in each round to make these types of attacks infeasible. That is because, based on OTS method, the sensors will have different cipher-text data in each round. Hence the attackers are not able to get the actual CS matrix as well as the information of the plaintext, resulting the CPA and KPA attacks to fail. However, OTS needs the sensor nodes and the BS to exchange the CS matrix (its size is much bigger than the CS

Original Image

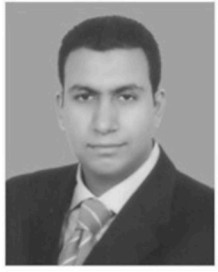

Reconstructed Image

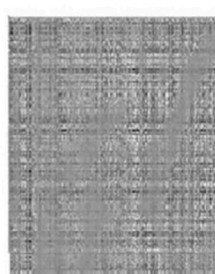

**Fig 6. Decryption process using tiny change in seed value.**

samples) in each round, which leads to increase the communication cost and reduce the network lifetime. As a solution to this issue, the proposed scheme uses a simple and efficient way to generate the CS encryption key (CS matrix) in each iteration without sharing any additional information between BS and the nodes as shown in section 4.8, where the sensors nodes and the BS use the CS global seed of the current round as the new initiation value for Eq 8 and generate a new seed separately without sharing any information. Therefore, it will be very difficult to launch an attack such as CPA and KPA to predict the CS matrix and the data is protected. Thus, the proposed system achieves the security goal and at the same time reduces the communication costs and prolongs the IoT network lifetime.

- Ciphertext-only attack (COA): In the COA attack model, the attacker is assumed to have access only to a set of ciphertexts. An adversary tries to figure out a plaintext by observing the corresponding ciphertext only. The security mechanism of the proposed security scheme encrypts the compressed samples at every node using a key (secret) value technique to ensure security. And if the COA attacker obtains the samples, he can only receive the encrypted version and not the original, making it impossible for him to achieve his goal. In addition, this secret value is a changeable value because the secret value depends on the global seed value that dynamically change in each round. Thus, increasing the security performance of the proposed scheme against COA.

Finally, during the inter- communication stage (from the CH to the BS) the proposed technique utilizes the public key algorithm which isn't affected by the statistical inference attacks discussed above. Furthermore, the private key is only used at the BS to decrypt data; there is no need to exchange it inside the network.

**6.1.2 Complexity analysis.** Table 2 gives a comparison of the proposed technique with other CS-based schemes in terms of different factors: Security technique, Attack model, CS OTS, and Encryption overload (which we calculate as the unencrypted text size divided by the encrypted text size). From Table 2, we can notice that the encryption overhead of our approach is equal to $M/(N − K) + k_s/K$, where $N$, $M$, $k$ and $k_s$ represents the nodes count, sample size, clusters count and ECC key size respectively.

In the CS schemes [33, 47–50], the Encryption overload is expressed as $M/N$. It is equal to $M/(N − k)$ for [47], and for Public key based schemes of [52] and [51], it is given by $Mk_H/N$, where $k_H$ denotes the key size of additive homomorphism algorithm adopted in them and $|R|/N+ 2M(log_2 q+ 1)/n$, where $q$, $R$ represents the prime power used for encryption and CipherText expansion, respectively.

**Table 2. Proposed scheme and other related approaches: A comparison.**

| Approach | Security technique | CS OTS | Attack Model | Encryption overhead |
|---|---|---|---|---|
| [49] | CS encryption | Not considered | Not proposed | $M/N$ |
| [48] | CS encryption | Not considered | Not proposed | $M/N$ |
| [50] | CS encryption | Not considered | Not proposed | $M/N$ |
| [42] | CS based encryption and Public Key | Not considered | Not Proposed | $M + k_H/N$ |
| [47] | CS encryption | Not considered | Not proposed | $M/N$ |
| [46] | CS encryption | Not considered | Not proposed | $M/(N − k)$ |
| [33] | CS encryption | Not considered | CPA | $M/N$ |
| [52] | Public Key | Not considered | CPA, KPA | $Mk_H/N$ |
| [51] | Public Key | Not Considered | CPA | $|R|/N + 2M(log_2 q + 1)/n$ |
| Proposed | CS based encryption and Public Key | Achieved | All | $M/(N − K) + k_s/K$ |

Finally, [42] used CS and Homomorphic public algorithm with $M + k_H/N$ Encryption overhead where, $k_H$ is Homomorphic key size. Based on the previous calculations, we may infer that the CS scheme has the lowest communication cost, but does not provide adequate performance with respect to security concerns. However, in the proposed scheme, the public key is only used by the CHs, rather than by all nodes as in other schemes [51, 52]. Because of this, the Encryption overhead of our technique is less than the others, which reduces the communication cost. As a result, the proposed technique outperforms the other CS schemes in terms of security. This demonstrates that the suggested technique has both good security and load balancing resulting from the use of a public-key based encryption method and the data size reduction through the use of CS.

## 6.2 Performance comparison

In this section, we provide the performance comparison between the proposed technique and CDG [34], LSS [33], EIREC [50], SDC [42] and EECSR [44] algorithms in terms of metrics: (i) energy efficiency, (ii) alive nodes count and (iii) network lifetime. We first discuss the simulation environment, followed by the performance evaluation in terms of the aforementioned metrics.

**6.2.1 Simulation settings.** The simulation settings used are same as that of EECSR [44] scheme, where $N = 100$ sensor nodes are deployed in the network region of size $100m \times 100m$. The BS position is the center of the network. For our technique, we used the PMLEACH algorithm [73] to organize the nodes into clusters, where N nodes are grouped into different clusters. The CMs of every cluster transfer their data to respective CHs. The CHs perform aggregation and transmit the resultant data to BS.

**6.2.2 Network lifetime performance.** Compared to other schemes, Fig 7 gives the effectiveness of the proposed technique in enhancing the lifetime of WSN with respect to first, half, and last node dead. That is because the proposed technique is based on the PMLEACH algorithm, which has an advantage over other routing protocols in terms of extending network lifetime. In addition, the proposed technique doesn't share huge information between the BS and nodes which has significant effect on the performance of PMLEACH.

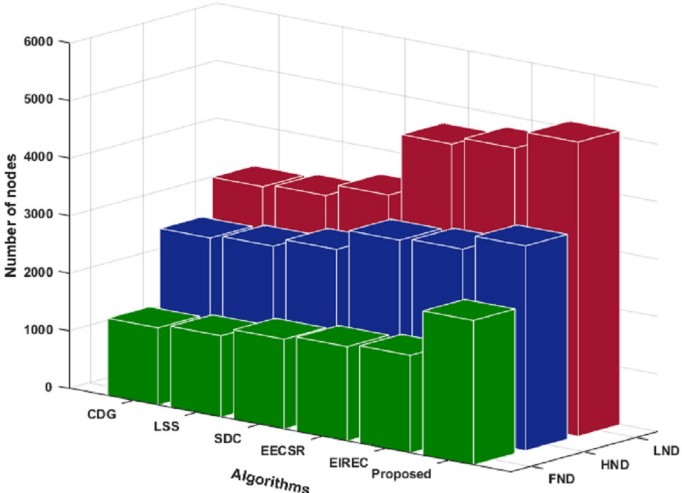

**Fig 7. FND, HND and LND results.**

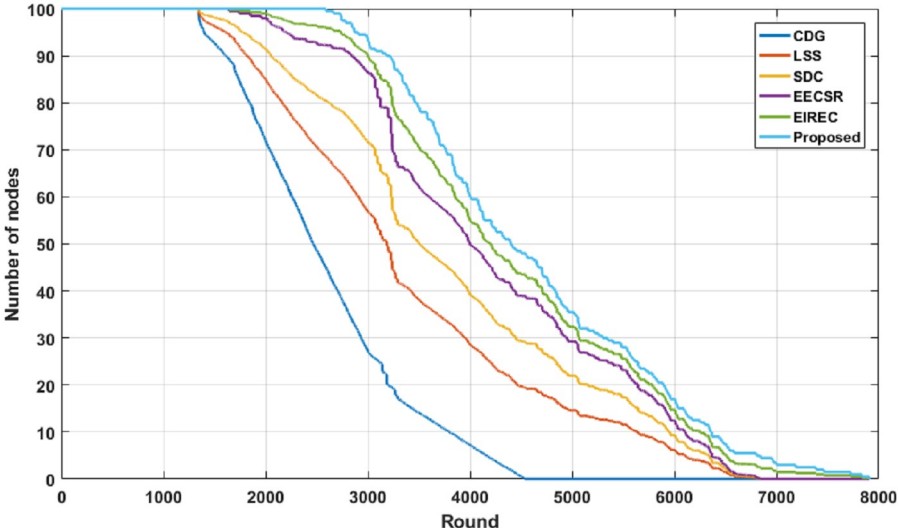

**Fig 8. The count of alive nodes versus round.**

**6.2.3 Alive nodes count.** We can find a comparison of the alive nodes count per round in Fig 8. From Fig 8, it is clear that the proposed solution lowers the count of dead nodes per round compared to the other schemes. The reason is that the suggested technique doesn't need to exchange the CS matrix in each round, but instead uses the proposed key re-generation process, which leads to the reduction of communication cost in each round and improves the lifetime of the network.

**6.2.4 Energy efficiency.** From Figs 9 and 10, we can recognize that the proposed strategy achieves the best results than the other schemes in minimizing the average energy expended. This is due to the fact that all complex computations (such as seed generation and CH selection) are transferred to the BS side, which has no energy constraint, resulting in enhanced

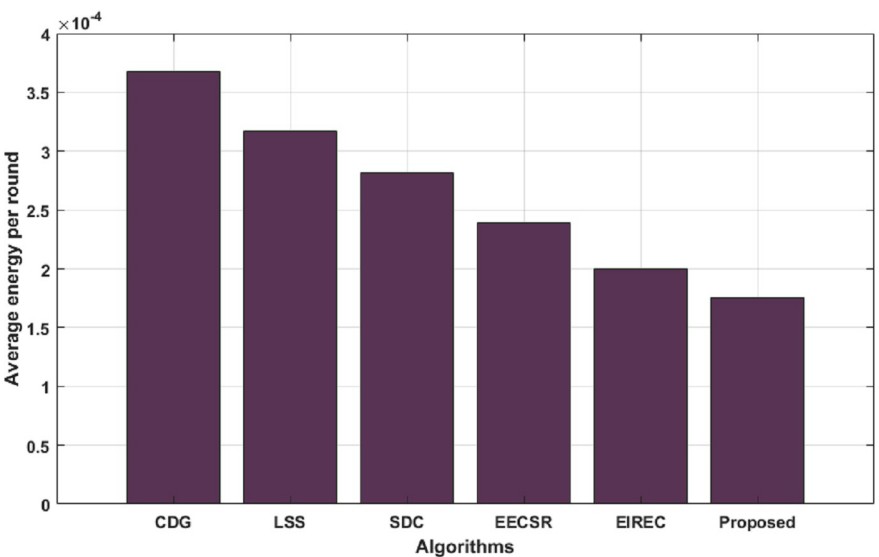

**Fig 9. Average energy consumed until FND.**

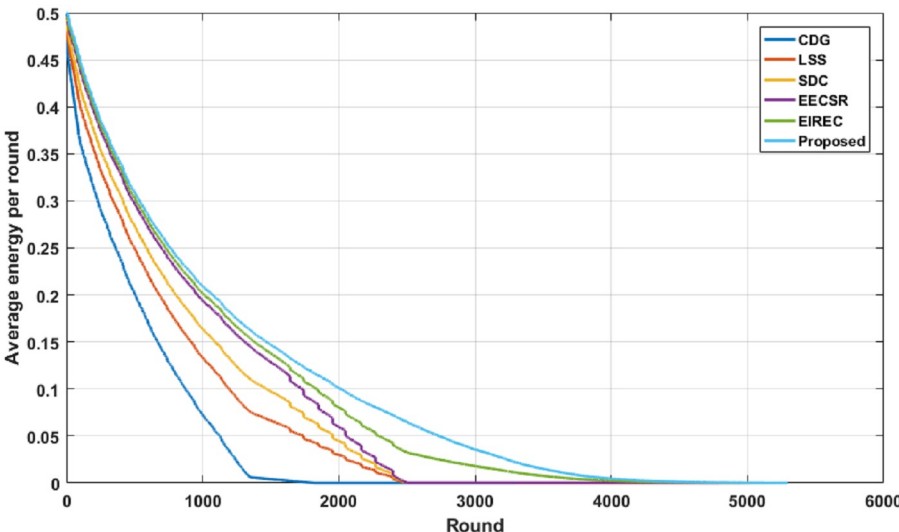

**Fig 10.  Average Energy of Network vs. Rounds.**

energy performance of the proposed scheme. Moreover, Fig 9 clearly indicates that the proposed solution still achieves the best performance than others. That's because the proposed system does not require extra memory to pre-load the secret key, which decreases the energy utilization and prolongs the lifetime of the WSN.

In summary, from the previous results, we can conclude that the proposed algorithm utilizes both CS-based encryption method and public key algorithm to accomplish high-security performance with less communication cost, that's achieved by using the CS scheme to encrypt and compress the sensor data through the data compression and CS-based encryption stage and Key sharing stage. In addition, to improve the security performance of this stage, the proposed scheme uses a efficient key sharing method and secret value technique which protects CS method from the different attacks model. Furthermore, the suggested scheme encrypts the cluster data using a public-key mechanism, preventing CS attacks at the data aggregation and ECC encryption phases. As a result, the proposed scheme outperforms the other CS schemes in terms of security and lifetime performance of WSN.

## 7 Conclusion

We have presented a security technique using the CS-based encryption/decryption method in combination with Elliptic Curve Cryptography (ECC) algorithm. This technique operates in five phases, namely, Key generation, CS-Key exchange, Data Compression with CS Encryption, Data Aggregation and Encryption with ECC algorithm and CS Key re-generation. The BS and the nodes adopt the use of two distinct chaotic maps to produce random numbers and seed. Besides this, the BS and the nodes securely exchange the seed in a simplified manner. Finally, the Compression with encryption concept produces secret compressed samples and offer protection against CPA, COA, and KPA attacks. The simulation results clearly depict that our technique can protect the CS matrix and prolong the WSN lifetime compared to existing algorithms.

## Acknowledgments

The authors gratefully acknowledge Qassim University, represented by the Deanship of Scientific Research, on the financial support for this research under the number (mcs- as-2020-1-3-I-10160) during the academic year 1441 AH /2020 AD.

## Author Contributions

**Conceptualization:** Ahmed Ismail, Walid Osamy.

**Data curation:** Ahmed Ismail, Walid Osamy.

**Formal analysis:** Ahmed Ismail, Walid Osamy.

**Funding acquisition:** Ahmed Ismail, Walid Osamy.

**Investigation:** Ahmed Salim, Ahmed Ismail, Walid Osamy.

**Methodology:** Ahmed Ismail.

**Project administration:** Ahmed Salim.

**Resources:** Ahmed Salim, Ahmed Ismail.

**Software:** Ahmed Salim, Walid Osamy, Ahmed M. Khedr.

**Supervision:** Ahmed Salim, Walid Osamy, Ahmed M. Khedr.

**Validation:** Ahmed Salim, Walid Osamy, Ahmed M. Khedr.

**Visualization:** Ahmed Salim, Ahmed M. Khedr.

**Writing – original draft:** Ahmed M. Khedr.

**Writing – review & editing:** Ahmed M. Khedr.

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
