## [Decision Letter · Decision Letter 0]

7 Jul 2021

PONE-D-21-19240

Compressive sensing based secure data aggregation scheme for IoT enabled WSNs applications

PLOS ONE

Dear Dr. Ismail,

Thank you for submitting your manuscript to PLOS ONE. After careful consideration, we feel that it has merit but does not fully meet PLOS ONE’s publication criteria as it currently stands. Therefore, we invite you to submit a revised version of the manuscript that addresses the points raised during the review process.

We look forward to receiving your revised manuscript.

Kind regards,

Pandi Vijayakumar, Ph.D

Academic Editor

PLOS ONE

Journal Requirements:

3. Please amend the manuscript submission data (via Edit Submission) to include author Ahmed Salim, Ahmed Aziz.

4. Please amend your authorship list in your manuscript file to include author Ahmed Ismail, Ahmed Salem.

5. We note that Figures 5 and 6 includes an image of a participant in the study. 

Reviewers' comments:

Reviewer's Responses to Questions

**Comments to the Author**

1. Is the manuscript technically sound, and do the data support the conclusions?

Reviewer #1: Partly

Reviewer #2: Yes

2. Has the statistical analysis been performed appropriately and rigorously? 

Reviewer #1: No

Reviewer #2: Yes

3. Have the authors made all data underlying the findings in their manuscript fully available?

Reviewer #1: Yes

Reviewer #2: Yes

4. Is the manuscript presented in an intelligible fashion and written in standard English?

Reviewer #1: Yes

Reviewer #2: Yes

5. Review Comments to the Author

Reviewer #1: The picture quality of the fig.3 is very poor. The authors should redraw the figure properly.

The novelty of the work is moderate.

The authors should give the brief explanation about the IoT enabled WSNs in the introduction. Why the authors keep the title as IoT enabled WSNs?

The authors should make formal security analysis.

Reviewer #2: Compressive sensing based secure data aggregation scheme for IoT enabled WSNs applications is presented in this paper. This paper has not clearly shown the advantages in performance of their approach with respect to others from the literature in this field. Indeed, I found the paper a little bit difficult to read, due not only to the poor grammar used throughout, but also the unclear structure of the argument being put across. In particular, the quality of the presentation should be improved in this paper. This paper would be substantially improved by thoroughly rewriting the prose with the help of a good English-language writer. In general, this paper needs such a treatment before being considered any further. Furthermore, presentation aside, by reading the paper, it still was not entirely clear what to expect with the direction of the article. Indeed, the contribution proposed in this paper should properly be compared and contextualized with respect to state of the art. The aspects mentioned above should be carefully addressed before the paper can be considered any further. Please consider the following remarks to improve your article:

Explain novelty of your work presented in this work.

Paper needs to polish and provide a detailed explication of theoretical aspects such as conditions and theorems, and practical issues like algorithms, rules and possible applications.

Introduction section needs to be re-written to improve its quality and readability.

Improve the quality of figures and explain those properly.

Following are some of relevant and recent references which need to be discussed in the revised manuscript:

IoT-based Big Data secure management in the Fog over a 6G Wireless Network.

Using Clustering for Forensics Analysis on Internet of Things

A multi-agent-based data collection and aggregation model for fog-enabled cloud monitoring

Security in Internet of Things: issues, challenges, taxonomy, and architecture

A Secure Decentralized Spatial Crowdsourcing Scheme for 6G-Enabled Network in Box

IoT transaction processing through cooperative concurrency control on fog–cloud computing environment

Many references are with incomplete bibliographic information (like lack of publication venue, for instance). This must be corrected

There are many English and grammatical issues in the paper which needs to be rectified.

The formula character format is best to be different from the main text, and mathematical characters are recommended.

In the related works, "et al" should be "et al.".

It seems that the contribution points of the article are a little bit few. After or in the section of Motivation, it is recommended that the authors summarize the contribution points of their work, which clearly demonstrate the innovations.

6. PLOS authors have the option to publish the peer review history of their article (what does this mean?). If published, this will include your full peer review and any attached files.

Reviewer #1: No

Reviewer #2: No

---

## [Author Response · Author response to Decision Letter 0]

8 Oct 2021

Compressive sensing based secure data aggregation scheme for IoT enabled WSNs applications

Revised title: Compressive sensing based secure data aggregation scheme for IoT Based WSN applications

Reviewers' comments

Reviewer #1

Question Answer

The picture quality of the fig.3 is very poor. Fixed, the figures are redrawn with better quality

The authors should redraw the figure properly. Fixed, the figures are redrawn with better quality

The novelty of the work is moderate. We have revised the paper to highlight the significance of the proposed scheme and create a separate subsection called Motivation and Contribution please see section 1.1

The authors should give the brief explanation about the IoT enabled WSNs in the introduction. The paper is revised for better understanding and the details about IoT enabled WSNs are provided in introduction: lines 5-12 

Why the authors keep the title as IoT enabled WSNs? now, and we changed it into IoT based WSNs instead of Fixed, the title is revised IoT enabled WSNs 

The authors should make formal security analysis. Thanks for the valuable comments, we have modified the security analysis section accordingly. Please see section 5.1 

Reviewer #1

Question Answer

This paper has not clearly shown the advantages in performance of their approach with respect to others from the literature in this field. Fixed the Related Work section is revised to highlight the benefits of our scheme. Moreover, a formal security analysis is added see section 5.1 to give bit understanding on the security feature of the proposed work. 

Indeed, I found the paper a little bit difficult to read, due not only to the poor grammar used throughout, but also the unclear structure of the argument being put across. Fixed, we have done a major revision of the language as well as the flow of the paper for better readability and understanding

In particular, the quality of the presentation should be improved in this paper. Fixed, we have done a major revision of the language as well as the flow of the paper for better readability and understanding

This paper would be substantially improved by thoroughly rewriting the prose with the help of a good English-language writer. Fixed, we have done a major revision of the language as well as the flow of the paper for better readability and understanding

In general, this paper needs such a treatment before being considered any further. Fixed, we have done a major revision of the language as well as the flow of the paper for better readability and understanding

Furthermore, presentation aside, by reading the paper, it still was not entirely clear what to expect with the direction of the article. Fixed, the paper is revised for better readability. Moreover, the significance of the proposed work are highlighted in section 1.1

Indeed, the contribution proposed in this paper should properly be compared and contextualized with respect to state of the art. Fixed the Related Work section is revised to highlight the benefits of our scheme. Moreover, a formal security analysis is added see section 5.1 to give bit understanding on the security feature of the proposed work. 

The aspects mentioned above should be carefully addressed before the paper can be considered any further. Fixed

Please consider the following remarks to improve your article:

Explain novelty of your work presented in this work. We have revised the paper to highlight the significance of the proposed scheme and create a separate subsection called Motivation and Contribution please see section 1.1

Following are some of relevant and recent references which need to be discussed in the revised manuscript::

1-IoT-based Big Data secure management in the Fog over a 6G Wireless network

2- Using Clustering for Forensics Analysis on Internet of Things

3- A multi-agent-based data collection and aggregation model for fog-enabled cloud monitoring

4- Security in Internet of Things: issues, challenges, taxonomy, and

Architecture

5- A Secure Decentralized Spatial Crowdsourcing Scheme for 6G-Enabled Network in Box

6- IoT transaction processing through cooperative concurrency control on

fog–cloud computing environment Thanks for the valuable comments, we have added the mentioned references please check the related work section. 

Paper needs to polish and provide a detailed explication of theoretical aspects such as conditions and theorems, and practical issues like algorithms, rules and possible applications.

 Fixed, we have done a major revision of the language as well as the flow of the paper for better readability and understanding

Introduction section needs to be re-written to improve its quality and readability. Fixed, this section is now revised to improve the readability and understanding 

Improve the quality of figures and explain those properly. Fixed, the figures are redrawn with better quality

Following are some of relevant and recent references which need to be discussed in the revised manuscript:

Many references are with incomplete bibliographic information (like lack of publication venue, for instance). This must be corrected. There are many English and grammatical issues in the paper which needs to be rectified. Fixed, we have done a major revision of the language as well as the flow of the paper for better readability and understanding

The formula character format is best to be different from the main text, and mathematical characters are recommended.

In the related works, "et al" should be "et al.". Fixed

It seems that the contribution points of the article are a little bit few. After or in the section of Motivation, it is recommended that the authors summarize the contribution points of their work, which clearly demonstrate the innovations. We have revised the paper to highlight the significance of the proposed scheme and create a separate subsection called Motivation and Contribution please see section 1.1

---

## [Decision Letter · Decision Letter 1]

22 Oct 2021

PONE-D-21-19240R1Compressive sensing based secure data aggregation scheme for IoT based WSNs applicationsPLOS ONE

Dear Dr. Ismail,

Thank you for submitting your manuscript to PLOS ONE. After careful consideration, we feel that it has merit but does not fully meet PLOS ONE’s publication criteria as it currently stands. Therefore, we invite you to submit a revised version of the manuscript that addresses the points raised during the review process.

We look forward to receiving your revised manuscript.

Kind regards,

Pandi Vijayakumar, Ph.D

Academic Editor

PLOS ONE

Journal Requirements:

Additional Editor Comments:

The authors should give more stress for novelty part.

Reviewers' comments:

Reviewer's Responses to Questions

**Comments to the Author**

1. If the authors have adequately addressed your comments raised in a previous round of review and you feel that this manuscript is now acceptable for publication, you may indicate that here to bypass the “Comments to the Author” section, enter your conflict of interest statement in the “Confidential to Editor” section, and submit your "Accept" recommendation.

Reviewer #1: (No Response)

Reviewer #2: All comments have been addressed

2. Is the manuscript technically sound, and do the data support the conclusions?

Reviewer #1: (No Response)

Reviewer #2: Yes

3. Has the statistical analysis been performed appropriately and rigorously? 

Reviewer #1: (No Response)

Reviewer #2: Yes

4. Have the authors made all data underlying the findings in their manuscript fully available?

Reviewer #1: (No Response)

Reviewer #2: Yes

5. Is the manuscript presented in an intelligible fashion and written in standard English?

Reviewer #1: (No Response)

Reviewer #2: Yes

6. Review Comments to the Author

Reviewer #1: The novelty of the work is limited. The following papers should be discussed in the related work section properly.

1.Comprehensive survey on security services in vehicular ad-hoc networks, IET Intelligent Transport Systems.

2. Dual authentication and key management techniques for secure data transmission in vehicular ad hoc networks

3.EAAP: Efficient anonymous authentication with conditional privacy-preserving scheme for vehicular ad hoc networks

4.Computationally efficient privacy preserving authentication and key distribution techniques for vehicular ad hoc networks

5.An Anonymous Batch Authentication and Key Exchange Protocols for 6G Enabled VANETs

6.BBAAS: Blockchain-Based Anonymous Authentication Scheme for Providing Secure Communication in VANETs

7.EMBA: An efficient anonymous mutual and batch authentication schemes for vanets.

Reviewer #2: Compressive sensing based secure data aggregation scheme for IoT based WSNs applications is presented in this paper and it is revised well.

7. PLOS authors have the option to publish the peer review history of their article (what does this mean?). If published, this will include your full peer review and any attached files.

Reviewer #1: No

Reviewer #2: No

---

## [Author Response · Author response to Decision Letter 1]

30 Oct 2021

Q1.The authors should give more stress for novelty part:

Answer:

We have revised the paper to highlight the significance of the proposed scheme, please see section 1.1

Q2. properly.

1.Comprehensive survey on security services in vehicular ad-hoc networks, IET Intelligent Transport Systems.

2. Dual authentication and key management techniques for secure data transmission in vehicular ad hoc networks

3.EAAP: Efficient anonymous authentication with conditional privacy-preserving scheme for vehicular ad hoc networks

4.Computationally efficient privacy preserving authentication and key distribution techniques for vehicular ad hoc networks

5.An Anonymous Batch Authentication and Key Exchange Protocols for 6G Enabled VANETs

6.BBAAS: Blockchain-Based Anonymous Authentication Scheme for Providing Secure Communication in VANETs

7.EMBA: An efficient anonymous mutual and batch authentication schemes for vanets.

Answer:

Thanks a lot for the suggestion, Fixed the Related Work section is revised and the proposed papers have been added and discussed.

---

## [Decision Letter · Decision Letter 2]

15 Nov 2021

Compressive sensing based secure data aggregation scheme for IoT based WSNs applications

PONE-D-21-19240R2

Dear Dr. Ismail,

We’re pleased to inform you that your manuscript has been judged scientifically suitable for publication and will be formally accepted for publication once it meets all outstanding technical requirements.

Kind regards,

Pandi Vijayakumar, Ph.D

Academic Editor

PLOS ONE

Additional Editor Comments (optional):

Both the reviewers have recommended the paper for acceptance. Hence, this paper can be accepted for publication.

Reviewers' comments:

Reviewer's Responses to Questions

**Comments to the Author**

1. If the authors have adequately addressed your comments raised in a previous round of review and you feel that this manuscript is now acceptable for publication, you may indicate that here to bypass the “Comments to the Author” section, enter your conflict of interest statement in the “Confidential to Editor” section, and submit your "Accept" recommendation.

Reviewer #1: All comments have been addressed

Reviewer #2: (No Response)

2. Is the manuscript technically sound, and do the data support the conclusions?

Reviewer #1: Yes

Reviewer #2: (No Response)

3. Has the statistical analysis been performed appropriately and rigorously? 

Reviewer #1: (No Response)

Reviewer #2: Yes

4. Have the authors made all data underlying the findings in their manuscript fully available?

Reviewer #1: (No Response)

Reviewer #2: Yes

5. Is the manuscript presented in an intelligible fashion and written in standard English?

Reviewer #1: (No Response)

Reviewer #2: Yes

6. Review Comments to the Author

Reviewer #1: (No Response)

Reviewer #2: Compressive sensing based secure data aggregation scheme for IoT based WSNs applications is presented in this paper. Paper is revised well. It can be accepted now.

7. PLOS authors have the option to publish the peer review history of their article (what does this mean?). If published, this will include your full peer review and any attached files.

Reviewer #1: No

Reviewer #2: No

---

## [Editor Report · Acceptance letter]

29 Nov 2021

PONE-D-21-19240R2 

Compressive sensing based secure data aggregation scheme for IoT Based WSN applications 

Dear Dr. Ismail:

I'm pleased to inform you that your manuscript has been deemed suitable for publication in PLOS ONE. Congratulations! Your manuscript is now with our production department. 

Kind regards, 

on behalf of

Dr. Pandi Vijayakumar 

Academic Editor

PLOS ONE